# Lightweight mobile network for real-time violence recognition

**Youshan Zhang**[1], **Yong Li**[2]*, **Shaozhe Guo**[1]

**1** Graduate Student Brigade, Chinese People's Armed Police Force Engineering University, Xi'an, Shaanxi Province, China, **2** College of Information Engineering, Chinese People's Armed Police Force Engineering University, Xi'an, Shaanxi Province, China

* liyong@nudt.edu.cn

## Abstract

Most existing violence recognition methods have complex network structures and high cost of computation and cannot meet the requirements of large-scale deployment. The purpose of this paper is to reduce the complexity of the model to realize the application of violence recognition on mobile intelligent terminals. To solve this problem, we propose MobileNet-TSM, a lightweight network, which uses MobileNet-V2 as main structure. By incorporating temporal shift modules (TSM), which can exchange information between frames, the capability of extracting dynamic characteristics between consecutive frames is strengthened. Extensive experiments are conducted to prove the validity of this method. Our proposed model has only 8.49MB parameters and 175.86MB estimated total size. Compared with the existing methods, this method greatly reduced the model size, at the cost of an accuracy gap of about 3%. The proposed model has achieved accuracy of 97.959%, 97.5% and 87.75% on three public datasets (Crowd Violence, Hockey Fights, and RWF-2000), respectively. Based on this, we also build a real-time violence recognition application on the Android terminal. The source code and trained models are available on https://github.com/1840210289/MobileNet-TSM.git.

## Introduction

With the development of Internet technology [1], an increasing number of digital cameras have been deployed in public places to monitor the order of public places or deployed in a private space to protect personal safety. Irrespective of where the digital camera is deployed, it must always identify the video manually. This is difficult to achieve because of the huge cost of monitoring all real-time pictures. In recent years, the application of deep learning in the field of computer vision has been widely studied. Deep learning model can be trained to extract feature information and classify pictures instead of human beings; this has increasingly attracted attention.

Image classification refers to classifying images by extracting different feature information hidden in pixels, which is a basic problem in the field of computer vision, and it is also the basis of complex visual tasks such as behavior recognition, object detection, and image cutting.

**Data Availability Statement:** All relevant data are within the paper.

**Funding:** This work was supported by the Basic Research Fund of the Engineering University of PAP (WJY202120) and the innovative research project on training objects of high-level scientific

and technological talents of PAP (ZZKY20222304). The funders had no role in study design, data collection and analysis, decision to publish, or preparation of the manuscript.

**Competing interests:** The authors have declared that no competing interests exist.

As a specific task of behavior recognition, violence recognition also attracted the attention of many scholars. Its development process is similar to that of behavior recognition. Generally, it can be divided into two stages: manually designed descriptors and deep learning methods.

Early violence recognition methods mainly extracted features using manually designed descriptors. Based on the sensitivity of human vision to features, special descriptors were constructed to represent violent behaviors in videos and then classified using different machine learning algorithms. The following research classified violence by judging the presence of blood or abnormal sounds such as gunshots in the video [1–3]. Special descriptors, such as VIolent Flows (ViF) [4, 5] and Oriented VIolent Flows (OViF) [6], were designed to extract the characteristics of violent behavior, such as large range of action, short occurrence time and large change in movement direction. ViF considered statistics of how flow-vector magnitudes change over time. OViF pointed out that ViF can't handle magnitudes with the same magnitude but opposite directions well. By depicting the information involving both of motion magnitudes and motion orientations, OViF achieved better performance. Scale Invariant Feature Transform (SIFT) [7], Histograms of Oriented Gradient (HOG) [8], and their variants [9–13] were also learned to identify violent behaviors.

Deep learning has shown its application prospects in computer vision tasks [14–17]. Many classic deep learning models such as VGGNET [18], YOLO [19], ResNet [20] and their variants have also achieved good performance in the field of violence recognition [21–24]. With the development of deep learning, the accuracy on the major public datasets has been constantly refreshed; meanwhile, the volume of the model has become increasingly larger. In particular, after vision transformer (ViT) [25] broke the application barrier in the field of computer vision, a series of research [26–28] based on transformer have showed its potential on many difficult tasks. With numerous parameters and calculations, advanced algorithms can only run in top-level hardware environments. This promoted the development of lightweight research of deep learning to obtain smaller and faster models while ensuring accuracy. The emergence of GhostNet [29], ShuffleNet [30] and MobileNet [31–33] enables edge devices to directly run deep learning models.

As a specific task of behavior recognition, violence recognition belongs to pattern classification in the field of computer vision. Its main task is to identify whether the picture contains violent behavior. Different from the task of behavior recognition, violence recognition must have a more clear and detailed division of human behavior. In recent years, scholars have been committed to the research of violence recognition from two perspectives.

First, a variety of algorithms are fused to meet the needs of violence recognition. Spatiotemporal attention modules and frame-grouping method [34] were proposed to build a violence detection system. By averaging the channels and grouping three consecutive channel-averaged images, short-term dynamics can be well simulated which is very helpful for violence identification. A deep multi-net (DMN) [35] architecture based on AlexNet and GoogleNet was proposed for violence recognition in videos, which used transfer learning to solve the problem of abrupt camera motion. By integrating Fusion Convolutional Neural Network (Fusion-CNN), spatio-temporal attention modules and Bi-directional Convolutional LSTMs (BiConvLSTM) [36], a two-stream architecture was proposed for violence recognition and localization in videos.

Second, the algorithm performance is improved by replacing a part of the basic framework. By changing the 3D convolutional layer to a depth-separable convolutional layer [37], this mix convolution block greatly reduces the computational cost. Ref [24] introduces a semi-supervised approach into pre-trained I3D, which can improve accuracy by removing redundant data and focusing on useful visual information. Ref [38] uses CNN-LSTM to extract visual and

auditory information simultaneously, then, a shared semantic subspace is constructed based on an autoencoder mapping model, which can fuse segment level features.

However, most of the existing studies focus on improving the accuracy of the algorithm, while ignoring the complexity of the algorithm. To meet the requirements of practical application, violence recognition must satisfy high accuracy requirements and speed of real-time recognition. Therefore, to solve the aforementioned problems, a lightweight network model MobileNet-TSM, which was proposed. Our contributions are summarized as follows:

1. We propose a lightweight network model MobileNet-TSM to meet the dual requirements of accuracy and real-time for violence recognition.

2. We integrate the idea of temporal shift module, such that the original model can better extract the characteristic information of violent behavior, and further improve the accuracy of violence recognition.

3. We conducted several experiments on three commonly used violence recognition datasets: Hockey Fights, Crowd Violence and RWF-2000. We compared the advantages and disadvantages of the proposed method with those of the most advanced method in terms of accuracy, model parameters, and model size.

4. We deploy the trained model on the Android terminal and design a mobile app to show the application prospect of the method proposed in this study.

## Related work

### Depthwise separable convolutions

Depthwise separable convolutions, mainly proposed by the Google research team, are the soul of Inception and MobileNet series. They solve the main problem of maintaining the accuracy of the network's classification, while completely reducing the computational cost and memory usage of the model. The lightweight network with smaller size and faster speed enables mobile intelligent terminals and embedded devices to run the neural network model. The main idea of depthwise separable convolutions is to use multiple small-sized convolution cores instead of large-sized cores to simplify the network complexity and reduce the number of parameters. The main method used in this study refers to the depthwise separable convolution of MobileNet series, and uses the combination of depthwise and pointwise revolution to build the basic module of the network. Fig 1 presents the schematic of depthwise and pointwise revolution.

A depthwise convolution sets a convolution kernel for each input channel, and each convolution kernel performs a convolution operation independently to an input channel, thus greatly reducing the amount of calculation and parameters of the model. Then, pointwise revolution integrates the characteristic information extracted by each convolution core. The

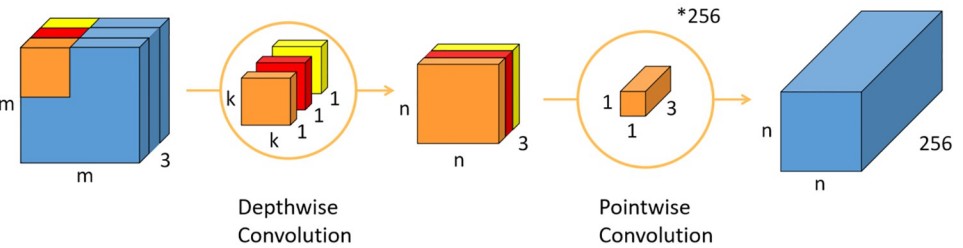

**Fig 1. Schematic of depthwise revolution and pointwise revolution.**

number of convolution cores is set equal to the number of output channels, and the size of convolution cores is $1 \times 1 \times N$, where N represents the number of input channels. The combined depthwise separable convolution replaces the traditional 3D convolution, thus reducing the amount of calculation by approximately 8 to 9 times.

## Temporal shift module

Temporal shift module (TSM) was proposed by Lin [39] in 2019, which is specially designed to exchange some position information between adjacent frames by shifting some channels in the time dimension before the convolution layer extracts the characteristic information. After that, 2D convolution kernel can extract spatial and temporal information at the same time, which can achieve the performance of 3D convolution neural network with the complexity of 2D convolution neural network. As shown in Fig 2(a), the original input tensor is stacked by several adjacent frames. We use different color to represent the input video frames at different times. As shown in Fig 2(b), by moving some channels along the temporal dimension in the same input batch, the information between frames is exchanged to a certain extent. When the following 2D convolution neural network is used to extract features, not only the spatial information of the current frame can be extracted, but also part of the temporal information of adjacent frames can be extracted.

For better expression, we assume that the input is an infinite one-dimensional vector $X^0$ and a 1 * 3 convolution kernel $W_1 = (a, b, c)$, then the convolution operation $Y = Conv(W, X^0)$ can be written as:

$$Y_i = a * x_{i-1} + b * x_i + c * x_{i+1} \tag{1}$$

$Y_i$ represents the i-th element in Y. $x_{i-1}$, $x_i$ and $x_{i+1}$ represent the (i-1)-th, i-th and (i+1)-th elements in the input vector X, respectively. $X^{-1}$, $X^{+1}$ represent the infinite one-dimensional vector $X^0$ shifted back and forth by a unit. The sum of $x_{i-1}$ is $X^{-1}$. The sum of $x_i$ is $X^0$. The sum of $x_{i+1}$ is $X^{+1}$. The shift operation can be written as:

$$X_i^{-1} = X_{i-1}, X_i^0 = X_i, X_i^{+1} = X_{i+1} \tag{2}$$

$$X^{-1} = \sum x_{i-1}, X^0 = \sum x_i, X^{+1} = \sum x_{i+1} \tag{3}$$

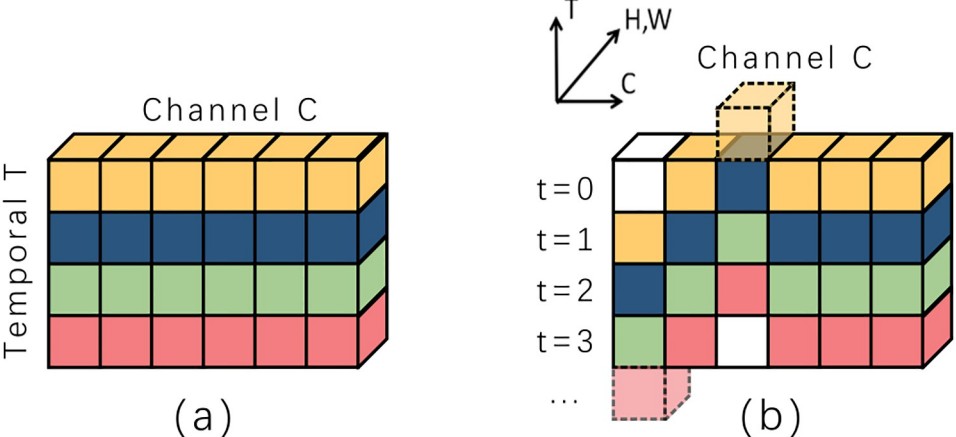

**Fig 2. (a) The original tensor without shift, (b) The tensor after shift.**

After that the convolution block followed will complete the multiply-accumulate operation as follows. Therefore, after passing through temporal shift module, each frame contains some information of adjacent frames.

$$Y = \sum Y_i = a * \sum x_{i-1} + b * \sum x_i + c * \sum x_{i+1} \qquad (4)$$

$$Y = a * X^{-1} + b * X^0 + c * X^{+1} \qquad (5)$$

## Partial temporal shift module

In our previous work, we proposed Partial temporal shift module [40] based on TSM [39]. We tried to improve the TSM algorithm from the perspectives of accuracy and real-time performance to meet the realistic requirements of violence recognition. By reducing temporal shift modules' insertion, our proposed optimal model had not only reduced the memory usage of hardware, but also achieved higher accuracy on multiple datasets with 77.3% running time. However, the P-TSM algorithm is still too complicated, so that it is still difficult to deploy applications in mobile intelligent terminals on a large scale.

## MobileNet-TSM

### Explanation

In order to deploy violence recognition applications on mobile intelligent terminals on a large scale, violence recognition significantly increases the demand for real-time and accuracy of the algorithm, especially the ability of fast recognition. Therefore, we turn our attention to the MobileNet series [31–33], which is a series of lightweight convolutional neural networks specially designed for mobile terminals or embedded devices. We carried out relevant experiments which inserted temporal shift module into MobileNet-V2 [32] and MobileNet-V3 [33], tried to further improve the performance of the model.

In this paragraph, we share two failed experiments to explain why we choose MobileNet-V2 as backbone structure. The first is to use MobileNet-V3 as the backbone structure, which exhibits the same accuracy as that of MobileNet-V2 on two smaller datasets, Hockey Fights and Crowd Violence. On RWF-2000, the highest accuracy we trained is 83.75% which is lower than that of MobileNet-V2. At the same time, MobileNet-V3 does not run as fast as Mobile-Net-V2 due to its higher model complexity. This runs counter to our initial goal of improvement. Another attempt is to further integrate the attention mechanism. We notice that the squeeze-and-excitation (SE) [41] attention module is integrated in MobileNet-V3 [33], but after inserting temporal shift modules, the model is always unable to fit. The same phenomenon also appears in the experiments combining the convolutional block attention module (CBAM) [42] in MobileNet-V2 [32]. We conjecture that the exchange of information between frames by the temporal shift module confuses the attention mechanism's feature extraction capability for key point regions. So we selects MobileNet-V2 [32] as the backbone after comparing the parameters and performance of the experiments.

### Main network structure

The proposed lightweight mobile network that can realize real-time violence recognition is mainly realized by inserting temporal shift modules in the backbone. The structure of the model is presented in Table 1. Conv2d represents two-dimensional convolution. Bottleneck is the main component of the model. Avgpool represents average pooling. Linear represents the full connection layer. t represents the expansion factor. c represents the number of output

**Table 1. Structure of MobileNet-TSM.**

| Input channel | Operation | t | c | n | s | m |
|---|---|---|---|---|---|---|
| $224^2 \times 3$ | Conv2d | 1 | 16 | 1 | 1 | - |
| $112^2 \times 32$ | Bottleneck | 6 | 24 | 2 | 2 | 1 |
| $112^2 \times 16$ | Bottleneck | 6 | 32 | 3 | 2 | 2 |
| $56^2 \times 24$ | Bottleneck | 6 | 64 | 4 | 2 | 3 |
| $28^2 \times 32$ | Bottleneck | 6 | 96 | 3 | 1 | 2 |
| $14^2 \times 64$ | Bottleneck | 6 | 160 | 3 | 2 | 2 |
| $7^2 \times 160$ | Bottleneck | 6 | 320 | 1 | 1 | - |
| $7^2 \times 320$ | Conv2d $1 \times 1$ | - | 1280 | 1 | 1 | - |
| $7^2 \times 1280$ | Avgpool $7 \times 7$ | - | - | 1 | - | - |
| $1 \times 1 \times 2$ | Linear | - | 2 | - | - | - |

channels. n represents that the corresponding bottleneck will repeat n times. s represents that the first layer of each part has a stride s and all others use stride 1. m represents the number of temporal shift modules we insert in the corresponding bottleneck.

In Fig 3, blocks with different color are used to represent different functional layers of the network. Bottleneck, which forms the main body of the network, mainly comprises several temporal shift modules inserted in the basis of MobileNet-V2 [32], as shown in Fig 4. The basic structure first uses pointwise revolution to upgrade the dimension of the input picture, with the expansion ratio of 6, and then uses temporal shift modules to exchange information among the input tensors. Batch normalization and ReLU6 are simultaneously used for normalization and activation. Then, depthwise revolution is used to extract the characteristic information of violent behaviors, and the combination of batch normalization and ReLU6 is also used. The projection layer, which is essentially a pointwise revolution, is used to compress data and tighten the network interface. In most of these bottleneck, shortcut connections are used to improve the performance of the model. Two additional issues must be addressed.

## Details

In this section, we explain two details of the network structure.

Why do we need to upgrade the dimension? If we downgrade the dimension of the input tensor, the calculation of the convolution layer will be reduced. The dilemma is that the higher the dimension of the input tensor, the more characteristic information can be extracted from

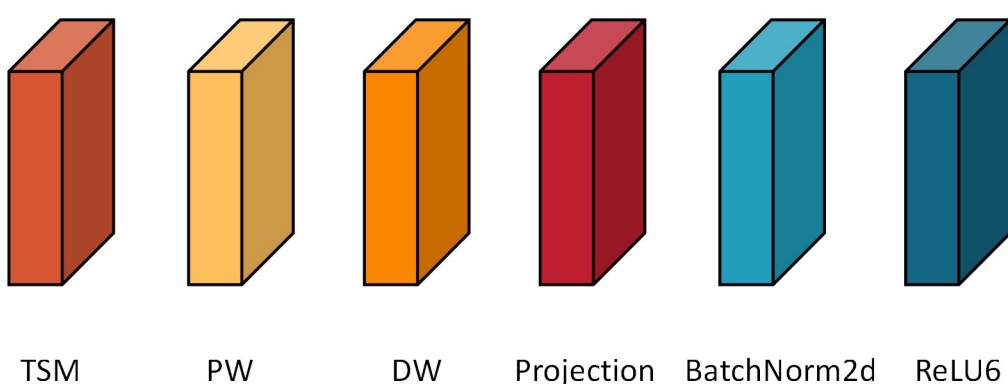

TSM　　　　PW　　　　DW　　　Projection　BatchNorm2d　ReLU6

**Fig 3. Diagram of basic module.**

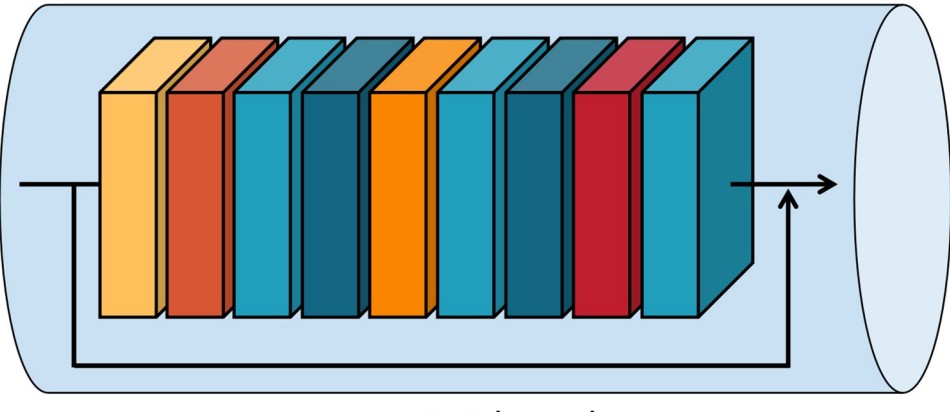

**Fig 4. Diagram of bottleneck structure.**

the convolution layer, and sufficient characteristic information can generate stable prediction results. Because the depth revolution can greatly reduce the amount of calculation, we can first use the pointwise revolution to upgrade the dimension, and then after the extraction the projection layer is used to downgrade the dimension. Both the pointwise revolution and projection layer have parameters to learn. Therefore, the entire network structure can learn how to better expand and compress the data.

Why do we insert the temporal shift modules in this way? The first reason is that we have considered the location of the temporal shift module. We insert the temporal shift module after the pointwise revolution instead of inserting it between each bottleneck similar to the original TSM algorithm. If we first perform temporal shift and then upgrade the dimension, the same information is extracted by six groups of convolution kernels. The information between adjacent frames exchanged by the temporal shift module will be diluted. The second reason is the performance of the model. Although one research [43] shows that appropriately increasing the number of temporal shift modules can improve the performance of the network, our experiments prove that increasing the number of temporal shift modules is not beneficial, particularly for models with a small capacity. Therefore, to maintain the stability of the model, the method we proposed only inserts the temporal shift module into the bottleneck that meets the following conditions: first, the stripe of the depthwise convolution is equal to 1. Second, the shape of the input characteristic matrix is the same as that of the output characteristic matrix. Third, shortcut connection is used.

## Experiments

### Datasets

In order to verify the performance of the proposed algorithm, we carried out experiments on three public violence recognition datasets. A summary of the used datasets is shown in Table 2 below.

The Hockey Fights [44] dataset contains 1000 violent and non-violent videos collected from ice hockey game. The training set includes 800 video clips, the verification set includes 100 video clips and the test set includes 100 video clips. The Crowd Violence [4] dataset mainly contains the scenes of crowd, but due to long shooting distance and low resolution, most of the scenes are chaotic and vague. The latest published RWF-2000 [45] dataset contains 2000 surveillance video clips collected from Youtube. The training set includes 1600 video clips, the

**Table 2. Introduction of used datasets for violence recognition.**

| Dataset | Year of release | Clips include | FPS | Frame resolution |
|---|---|---|---|---|
| Crowd Violence [4] | 2012 | 246 | 25 | 320 × 240 |
| Hockey Fights [44] | 2011 | 1000 | 25 | 360 × 288 |
| RWF-2000 [45] | 2021 | 2000 | 30 | 300 × 240, 320 × 240, 480 × 360, 920 × 720, 1280 × 720 |

verification set includes 200 video clips and the test set includes 200 video clips. Each video clip is 5 seconds and contains 150 frames. It mainly includes violent behaviors such as two persons, multiple persons, and crowds. The scenes are so rich and complicated that it is difficult to recognize. All video clips are obtained through the security camera. Without multimedia technology transformation, they fit the actual scene and have high research value.

## Experimental environment and configuration

We rented a virtual computer on AI-Galaxy GPU cloud for experiments and using Windows 10 as the operating system. During the entire experiments, we used Pytorch 1.9.0 as the deep learning framework and Intel(R) Xeon(R) Gold 6130 @2.10GHz as the CPU. We use CUDA 11.1 to accelerate the GPU and two NVIDIA RTX 2080 Ti GPUs with 11 GB video memory for parallel computing.

During the experiment, we used the pre-trained model of MobileNet-V2 provided by PyTorch to reduce the computational complexity of network training. We set the initial learning rate as 0.000375 and used Adam optimizer. The momentum decay rate used is 0.9. The input modality of the picture is RGB. The batch size of the input picture is 8. The shift division of the temporal shift module is also set to 8. During training, when the curve of the model on the test set significantly fluctuated, we set the initial learning rate as 0.0005 or 0.0001 and change the learning rate adjustment method. We adjusted the learning rate to 85% of the original learning rate every two epochs to train a better model.

## Ablation experiment

**Accuracy of TSM module insertion experiments.** First, we first tested the accuracy of the baseline model, namely MobileNet-V2 [32], on three public datasets of violent behaviors, and then, we tested the model by inserting temporal shift modules as original TSM [39] (Baseline + TSM #x2212; 1). We inserted and conducted experiments as we designed (Baseline + TSM − 2). The experiments' accuracy is presented in Table 3.

The baseline model using method 2 to insert the temporal shift module achieved 97.959% accuracy on the Crowd Violence, 97.5% on the Hockey Fights, and 87.75% on the RWF-2000. Compared with the baseline model, the accuracy increased by 5.102%, 2.5%, and 4.75% respectively. The baseline model using method 1 to insert the temporal shift module also had a certain improvement in accuracy compared with that of the baseline model, but the performance was not as good as that of method 2. The experiments showed that the addition of the temporal

**Table 3. Introduction of used datasets for violence recognition.**

| | Crowd Violence | Hockey Fights | RWF-2000 |
|---|---|---|---|
| Baseline | 92.857% | 95% | 83% |
| Baseline+TSM-1 | 95.95% | 97.2% | 87% |
| Baseline+TSM-2 | 97.959% | 97.5% | 87.75% |

shift module could effectively improve the baseline's capability of extracting spatio-temporal information. Through cross-channel information interaction, the bottleneck of the model was made more sensitive to the temporal information of violent behaviors, thereby improving the recognition accuracy.

**Comparison of optimal methods.** To further compare the performance of our model with that of the state-of-the-art behavior recognition algorithm, we tested it on different datasets in the same experimental environment. Table 4 presents the specific performance of different algorithms.

As shown in the table, the proposed model, which inserts the temporal shift module, achieves similar or even better results than those of the other state-of-the-art behavior recognition algorithms on three public violence recognition datasets. The proposed method achieves 97.959% accuracy on the Crowd Violence which is higher than most of the algorithms. Only 0.191% lower than SAM, which achieved the best performance on the Crowd Violence. On Hockey Fights, our methods achieved 97.5% accuracy that was slightly lower than that of the two-cascade TSM [48]. There is a gap of 2% compared with SAM that obtains the optimal accuracy. On RWF-2000, the method only achieved 87.75% accuracy which is 3.25% lower than P-TSM. In terms of accuracy, our proposed method is still inferior to the state-of-the-art algorithms.

MiNet-3D [37] also proposed a lightweight model for violence recognition, which uses a combination of 3D convolution and 2D convolution as the basic framework. The model was lightened by using depthwise separable convolutions instead of 3D convolutions. The method we proposed uses MobileNet-V2 as the basic framework and integrates temporal shift module to improve the performance of the model. MobileNet-V2 uses deep separable convolution as its main structure.

As shown in Fig 5, we can see that the accuracy does not significantly improve, but our model has less parameters and lower computational complexity. The biggest advantage of our proposed method lies in the complexity of the model, which is discussed in detail in the next paragraph.

**Comparison of the computational cost.** To better demonstrate the advantages of the proposed method, we tested more related data to further compare the performance of the algorithms. We compared the computational cost of the proposed method with that of the existing algorithms, as presented in Table 5. Fig 6 shows the comparison of estimated total size.

**Table 4. Comparison of optimal accuracy of different algorithms.**

| Algorithm | Crowd Violence | Hockey Fights | RWF-2000 |
|---|---|---|---|
| Resnet-50 [20] | 93.878% | 95.5% | 84% |
| 3D-CNN [46] | 94.3% | 94.4% | 82.75% |
| LRCN [47] | 94.57% | 97.1% | 77% |
| I3D [48] | 88.89% | 97.5% | 85.75% |
| MiNet-3D [37] | 91.41% | 94.71% | 81.98% |
| AR-Net [49] | 95.918% | 97.2% | 87.3% |
| TSM [39] | 95.95% | 97.5% | 88% |
| TEA [50] | 96.939% | 97.7% | 88.5% |
| Two-cascade TSM [43] | 96.939% | 98.05% | 89% |
| SAM [36] | **98.15%** | **99.1%** | 89.1% |
| SSHA [24] | - | 98.7% | 90.4% |
| P-TSM [40] | 96.939% | 98.5% | **91%** |
| Baseline+TSM-2(ours) | 97.959% | 97.5% | 87.75% |

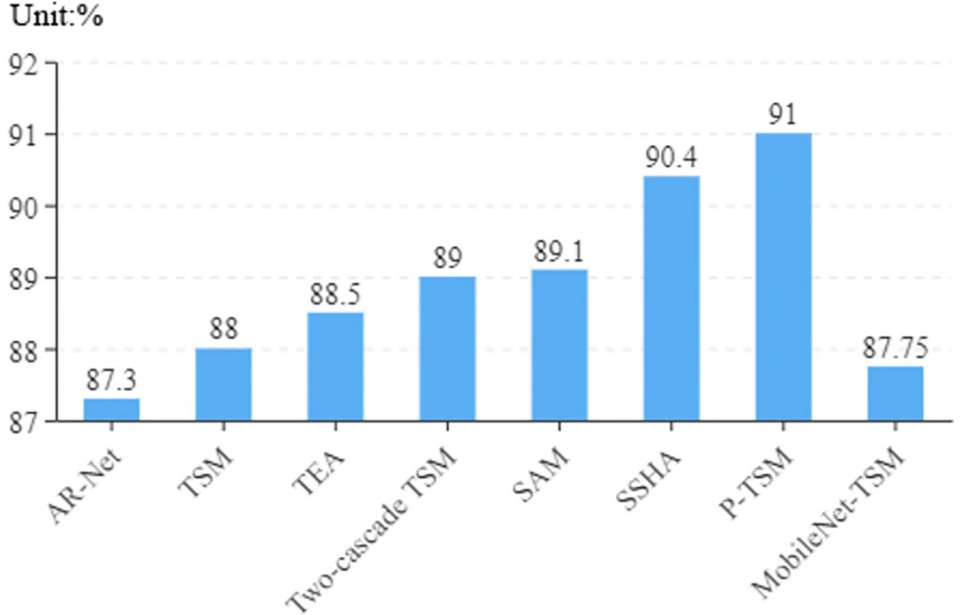

**Fig 5. Accuracy of different algorithms on RWF-2000 dataset.**

We used torchsummary to calculate models' parameters and estimated total size. Since torchsummary can control all variables of model computational complexity, and the calculation results can be simply reviewed instead of manual calculation. For better comparison, we set the input of the summary function is set to a fixed value. We set the picture size as $224 \times 224$, batch size as 8, and number of input channels as 3. We used Tensorboard to record relevant data that provides the time used after training and testing for 100 epochs.

As observed from the Table 5, compared with the existing behavior recognition algorithms, including the latest violence recognition algorithms, the proposed method significantly reduced the complexity of the model. The use of depthwise separable convolution considerably reduced the number of parameters to be trained. Our proposed model has only 8.49MB parameters and 175.86MB estimated total size. Compared with the state-of-the-art violence recognition algorithm, our proposed model reduces the used parameters by an order of magnitude. Furthermore, the training time was substantially reduced, and only takes 32 minutes and 41 seconds, which is a quarter of the time two-cascade TSM [43] taken to complete one

**Table 5. Complexity of different models.**

| Algorithm | Params | Estimated total size | Time cost |
|---|---|---|---|
| 3D-CNN [46] | 297.56MB | 2647.70MB | 4h8m23s |
| LRCN [47] | 237.83MB | 1212.93MB | 3h5m7s |
| I3D [48] | 46.88MB | 1000.20MB | 2h3m48s |
| TSN [51] | 89.69MB | 390.05MB | 46m41s |
| TSM [39] | 89.69MB | 397.71MB | 1h53m26s |
| TEA [50] | 91.95MB | 479.78MB | 3h10m48s |
| Two-cascade TSM [43] | 89.69MB | 397.71MB | 2h8m20s |
| P-TSM [40] | 89.69MB | 396.57MB | 1h27m38s |
| Baseline+TSM-2(ours) | **8.49MB** | **175.86MB** | **32m41s** |

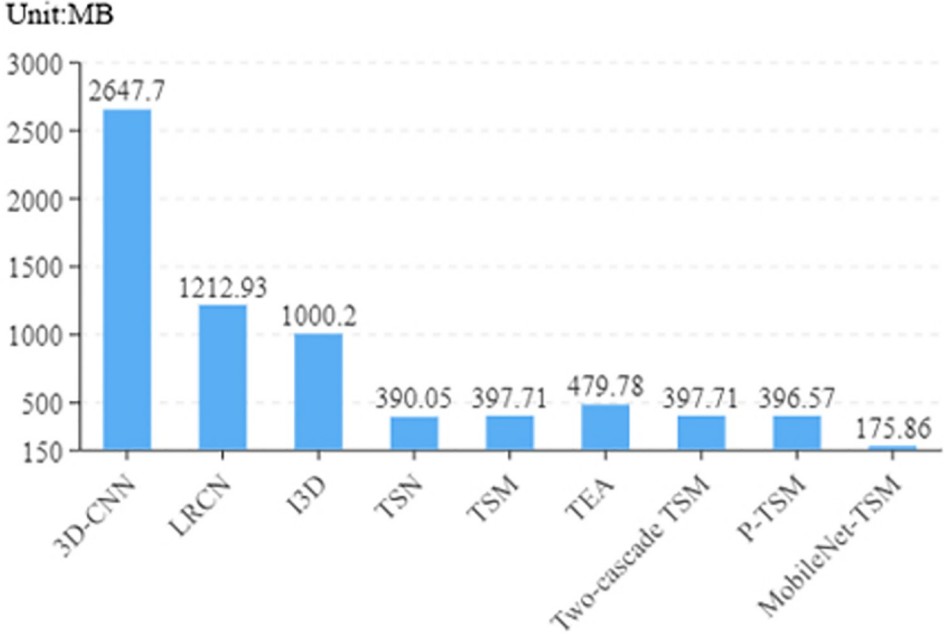

**Fig 6. Comparison of estimated total size.**

hundred epochs of training and testing. Fig 6 shows that the estimated total size of the model also greatly reduced, providing favorable conditions for the deployment in mobile intelligent terminals.

**Android smartphone implementation.** We used the Xiaomi MIX2S smartphone, Android system version 10, and Qualcomm Snapdragon 845 processor. It is worth mentioning that the android building Gradle used is 3.5.0, and the versions of "pytorch-android-lite" and "pytorch-android-torchvision" had to be the same as the PyTorch version when the deep learning model is training. Due to compatibility issues, we can only use PyTorch version 1.9.0 to do our work.

To deploy our trained model on the Android terminal, we first have to convert the format of the model. Built-in functions provided by Pytorch 1.9.0, optimize-for-mobile and save-for-lite-interpreter, were used to convert the format from.ph to.ptl. On the Android terminal, we can directly call the converted model.

The Android app mainly implemented two simple functions to demonstrate the practical performance of the proposed method. Function 1: Identify the existence of a violent behavior in the picture. After clicking the function button, the picture saved in the machine was read, converted into a tensor, and sent into the pre-trained deep learning model for prediction. Subsequently, the prediction result was output. Function 2: Identify the real-time image captured by the smartphone. After clicking the real-time detection function button, the app called the camera of the terminal to capture the real-time image, converted it into a tensor, and then input it into the pre-trained model for real-time detection and output prediction. Simultaneously, the current delay was showed.

## Conclusion

To solve the problem of huge computational cost of the existing violence recognition algorithm, in particular, not meeting the needs of real-time mobile terminal recognition, this

paper proposes a lightweight network called MobileNet-TSM. The experimental results on three public datasets, that is, Hockey Fights, Crowd Violence, and RWF-2000 showed that the proposed method significantly reduced the model complexity compared with the existing violence recognition algorithms. It could achieve a performance similar to that of the state-of-art algorithm with a relatively less number of training parameters and transplant it to smart mobile devices, verifying the feasibility of its application and showing a broad application prospect. In the future research of real-time violence recognition on mobile intelligent terminal, compatibility is a problem that needs to be solved including the deployment of deep learning models and mobile intelligent terminals with different operating systems.

## Author Contributions

**Investigation:** Yong Li.

**Methodology:** Youshan Zhang.

**Resources:** Yong Li.

**Software:** Youshan Zhang.

**Writing – original draft:** Youshan Zhang.

**Writing – review & editing:** Shaozhe Guo.

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
