## [Decision Letter · Decision Letter 0]

4 Sep 2022

PONE-D-22-21672Lightweight mobile network for real-time violence recognitionPLOS ONE

Dear Dr. Zhang,

Thank you for submitting your manuscript to PLOS ONE. After careful consideration, we feel that it has merit but does not fully meet PLOS ONE’s publication criteria as it currently stands. Therefore, we invite you to submit a revised version of the manuscript that addresses the points raised during the review process.

ACADEMIC EDITOR:The research proposes a MobileNet-TSM, a lightweight network for real time voice recognition. It has an acceptable structure and the experiments seems enough to show the performance. However, there are some comments to be considered before possible publication.

1-Explanation and justification for the use of MobileNet V2 as the backbone structure to be provided.

2-The abstract need to be improved, more about the advantage and disadvantages of your approach, novelty, and valuable results you observed.

3-For literature review, further discussions on these hot issues, you should supplement some newest references to papers published more recently, which appeared from 2021 on.

4-Every equation should be explained more clearly.

5-Table 4 and Table 5 results should be interpreted in detail.

We look forward to receiving your revised manuscript.

Kind regards,

Rajmohan Rajendran, Ph.D.

Academic Editor

PLOS ONE

Journal Requirements:

Additionally, please note that PLOS ONE has specific guidelines on software sharing (http://journals.plos.org/plosone/s/materials-and-software-sharing#loc-sharing-software) for manuscripts whose main purpose is the description of a new software or software package. In this case, new software must conform to the Open Source Definition (https://opensource.org/docs/osd) and be deposited in an open software archive. Please see http://journals.plos.org/plosone/s/materials-and-software-sharing#loc-depositing-software for more information on depositing your software.

"NO. The funders had no role in study design, data collection and analysis, decision to publish, or preparation of the manuscript."

Reviewers' comments:

Reviewer's Responses to Questions

**Comments to the Author**

1. Is the manuscript technically sound, and do the data support the conclusions?

Reviewer #1: Yes

Reviewer #2: Partly

Reviewer #3: Yes

2. Has the statistical analysis been performed appropriately and rigorously? 

Reviewer #1: Yes

Reviewer #2: Yes

Reviewer #3: Yes

3. Have the authors made all data underlying the findings in their manuscript fully available?

Reviewer #1: Yes

Reviewer #2: Yes

Reviewer #3: Yes

4. Is the manuscript presented in an intelligible fashion and written in standard English?

Reviewer #1: Yes

Reviewer #2: Yes

Reviewer #3: Yes

5. Review Comments to the Author

Reviewer #1: Review of the manuscript titled: Lightweight mobile network for real-time violence recognition

1. The objectives need to be well stated in the abstract

2. Performance metrics were not stated

3. The summary of the results should be stated in the abstract. Extensive experiments were conducted on three public datasets (Hockey Fights, Crowd Violence, and RWF-2000) to prove the validity of this method. What are the results of the experiment?

4. At least two more Keywords should be added

5. What is the trade-off of this proposed method?

6. What is the difference between this paper and A Lightweight Network for Violence Detection, ICIGP 2022: 2022 the 5th International Conference on Image and Graphics Processing (ICIGP)January 2022 Pages 15–21 https://doi.org/10.1145/3512388.3512391 ?

7. What informed the consideration of the deep separable convolution as the initial bottleneck.

8. The research gap in this study was not well established.

Reviewer #2: Introduction can be re framed including the objectives of the article.

Literature survey is not adequate. Many new papers on machine learning/deep learning algorithms are not included.

proposed method(MobileNet-TSM) is not explained well. Methodology is not adequate.

chosen Parameters are very low for analyzing.

Incorporate the comparison graph.

Reviewer #3: 1. In Page 2, Line 22, Can you supply more explanation to Oriented VIolent Flows (OViF) [6] ?

2. In Page 3, Line 57, how do you exactly deploy the trained model on the Android terminal ?

3. In Page 7, Line 222, why do you use torchsummary to calculate models’ parameters ?

4. In Page 8, Line 242, Why cant you use latest version instead of PyTorch version 1.9.0 ?

5. Can you give the scope for future enhancement of this research work ?

---

## [Author Response · Author response to Decision Letter 0]

10 Sep 2022

Dear editors and reviewers,

Thank you very much for your nice consideration of our manuscript and kind help with the reviewing of the manuscript entitled “Lightweight mobile network for real-time violence recognition” (Manuscript Number: PONE-D-22-21672 ). Also, we highly appreciate that the reviewers and editors gave some constructive and objective comments to this manuscript. Those comments are very helpful for revising and improving our manuscript. Here we submit a new version of the manuscript, which has been carefully revised according to comments of the reviewers and editors. All changes are highlighted in blue in the revised manuscript. Below is our detailed response to all of the reviewer’s comments.

---

## [Decision Letter · Decision Letter 1]

27 Sep 2022

PONE-D-22-21672R1Lightweight mobile network for real-time violence recognitionPLOS ONE

Dear Dr. Zhang,

Thank you for submitting your manuscript to PLOS ONE. After careful consideration, we feel that it has merit but does not fully meet PLOS ONE’s publication criteria as it currently stands. Therefore, we invite you to submit a revised version of the manuscript that addresses the points raised during the review process.

My comments, and any reviewer comments, are below.

We appreciate you submitting your manuscript to PLOS ONE and thank you for giving us the opportunity to consider your work.

We look forward to receiving your revised manuscript.

Kind regards,

Rajmohan Rajendran, Ph.D.

Academic Editor

PLOS ONE

Journal Requirements:

Additional Editor Comments:

The quality of the images has to be drastically improved for further progress of the article.

Reviewers' comments:

Reviewer's Responses to Questions

**Comments to the Author**

1. If the authors have adequately addressed your comments raised in a previous round of review and you feel that this manuscript is now acceptable for publication, you may indicate that here to bypass the “Comments to the Author” section, enter your conflict of interest statement in the “Confidential to Editor” section, and submit your "Accept" recommendation.

Reviewer #1: All comments have been addressed

Reviewer #2: All comments have been addressed

Reviewer #3: All comments have been addressed

2. Is the manuscript technically sound, and do the data support the conclusions?

Reviewer #1: Yes

Reviewer #2: Yes

Reviewer #3: Yes

3. Has the statistical analysis been performed appropriately and rigorously? 

Reviewer #1: Yes

Reviewer #2: Yes

Reviewer #3: Yes

4. Have the authors made all data underlying the findings in their manuscript fully available?

Reviewer #1: Yes

Reviewer #2: Yes

Reviewer #3: Yes

5. Is the manuscript presented in an intelligible fashion and written in standard English?

Reviewer #1: Yes

Reviewer #2: Yes

Reviewer #3: Yes

6. Review Comments to the Author

Reviewer #1: (No Response)

Reviewer #2: Image Quality must be enhanced. (At least 300 dpi). Fig.5 and Fig.6 originality is still not adequate. Comparison graphs shall be added.

Reviewer #3: All comments have been addressed by the authors.

---

## [Author Response · Author response to Decision Letter 1]

29 Sep 2022

Dear editors and reviewers,

Thank you very much for your nice consideration of our manuscript and kind help with the reviewing of the manuscript entitled “Lightweight mobile network for real-time violence recognition” (Manuscript Number: PONE-D-22-21672R1 ). Also, we highly appreciate that the reviewers and editors gave some constructive and objective comments to this manuscript. Those comments are very helpful for revising and improving our manuscript. Here we submit a new version of the manuscript, which has been carefully revised according to comments of the reviewers and editors. All changes are highlighted in blue in the revised manuscript. Below is our detailed response to all of the reviewer’s comments.

Comments of ACADEMIC EDITOR:

The quality of the images has to be drastically improved for further progress of the article.

Response: 

Thank you very much for your previous comments. Those comments had helped us a lot for improving our manuscript. All pictures have been redrawn. Preflight Analysis and Conversion Engine (PACE) digital diagnostic tool have been used to ensure figures meet PLOS requirements.

Comments of Reviewer 1:

(No Response)

Response: 

Thank you very much for your previous comments. Those comments had helped us a lot for improving our manuscript.

Comments of Reviewer 2:

Image Quality must be enhanced. (At least 300 dpi).

Fig.5 and Fig.6 originality is still not adequate.

Comparison graphs shall be added.

Response: 

Thank you very much for your previous comments. Those comments had helped us a lot for improving our manuscript.

Comment 1:

Image Quality must be enhanced. (At least 300 dpi).

Response: 

Thank you for your advice. All pictures have been redrawn. Preflight Analysis and Conversion Engine (PACE) digital diagnostic tool have been used to ensure figures meet PLOS requirements.

Comment 2:

Fig.5 and Fig.6 originality is still not adequate..

Response: 

Thank you for your advice. After our team's discussion, we thought that Figure 5 and Figure 6 only played a role of demonstration, so we decided to delete these two pictures.

Comment 3:

Comparison graphs shall be added.

Response: 

Thank you for your advice. Next to Table 4 and Table 5, we selected important performance indicators and made comparison graphs.

Comments of Reviewer 3: 

All comments have been addressed by the authors.

Response: 

Thank you very much for your previous comments. Those comments had helped us a lot for improving our manuscript.

---

## [Decision Letter · Decision Letter 2]

18 Oct 2022

Lightweight mobile network for real-time violence recognition

PONE-D-22-21672R2

Dear Dr. Zhang,

We’re pleased to inform you that your manuscript has been judged scientifically suitable for publication and will be formally accepted for publication once it meets all outstanding technical requirements.

Kind regards,

Rajmohan Rajendran, Ph.D.

Academic Editor

PLOS ONE

Additional Editor Comments (optional):

Reviewers' comments:

Reviewer's Responses to Questions

**Comments to the Author**

1. If the authors have adequately addressed your comments raised in a previous round of review and you feel that this manuscript is now acceptable for publication, you may indicate that here to bypass the “Comments to the Author” section, enter your conflict of interest statement in the “Confidential to Editor” section, and submit your "Accept" recommendation.

Reviewer #2: All comments have been addressed

2. Is the manuscript technically sound, and do the data support the conclusions?

Reviewer #2: Yes

3. Has the statistical analysis been performed appropriately and rigorously? 

Reviewer #2: Yes

4. Have the authors made all data underlying the findings in their manuscript fully available?

Reviewer #2: Yes

5. Is the manuscript presented in an intelligible fashion and written in standard English?

Reviewer #2: Yes

6. Review Comments to the Author

Reviewer #2: The authors have built a real-time violence recognition application. The authors have addressed all the comments.

7. PLOS authors have the option to publish the peer review history of their article (what does this mean?). If published, this will include your full peer review and any attached files.

Reviewer #2: **Yes: **Ananth kumar T

---

## [Editor Report · Acceptance letter]

20 Oct 2022

PONE-D-22-21672R2 

Lightweight mobile network for real-time violence recognition 

Dear Dr. Zhang:

I'm pleased to inform you that your manuscript has been deemed suitable for publication in PLOS ONE. Congratulations! Your manuscript is now with our production department. 

Kind regards, 

on behalf of

Dr. Rajmohan Rajendran 

Academic Editor

PLOS ONE